# SARS-CoV-2 Accessory Protein ORF8 Decreases Antibody-Dependent Cellular Cytotoxicity

**DOI:** 10.3390/v14061237

**Published:** 2022-06-07

**Authors:** Guillaume Beaudoin-Bussières, Ariana Arduini, Catherine Bourassa, Halima Medjahed, Gabrielle Gendron-Lepage, Jonathan Richard, Qinghua Pan, Zhen Wang, Chen Liang, Andrés Finzi

**Affiliations:** 1Centre de Recherche du CHUM, Montreal, QC H2X 0A9, Canada; guillaume.beaudoin-bussieres@umontreal.ca (G.B.-B.); catherine.bourassa.chum@ssss.gouv.qc.ca (C.B.); halima.medjahed.chum@ssss.gouv.qc.ca (H.M.); gabrielle.gendron-lepage.chum@ssss.gouv.qc.ca (G.G.-L.); jonathan.richard.1@umontreal.ca (J.R.); 2Département de Microbiologie, Infectiologie et Immunologie, Université de Montréal, Montreal, QC H3C 3J7, Canada; 3Lady Davis Institute, Jewish General Hospital, Montreal, QC H3T 1E2, Canada; ariana.arduini@mail.mcgill.ca (A.A.); panqinghua07@gmail.com (Q.P.); wangzhen092014@gmail.com (Z.W.); 4Department of Medicine, McGill University, Montreal, QC H3G 2M1, Canada; 5Department of Microbiology and Immunology, McGill University, Montreal, QC H3A 0G4, Canada

**Keywords:** coronavirus, SARS-CoV-2, ORF8, accessory protein, Fc-mediated effector function, ADCC, monocytes, NK cells, CD16

## Abstract

Viruses use many different strategies to evade host immune responses. In the case of SARS-CoV-2, its Spike mutates rapidly to escape from neutralizing antibodies. In addition to this strategy, ORF8, a small accessory protein encoded by SARS-CoV-2, helps immune evasion by reducing the susceptibility of SARS-CoV-2-infected cells to the cytotoxic CD8+ T cell response. Interestingly, among all accessory proteins, ORF8 is rapidly evolving and a deletion in this protein has been linked to milder disease. Here, we studied the effect of ORF8 on peripheral blood mononuclear cells (PBMC). Specifically, we found that ORF8 can bind monocytes as well as NK cells. Strikingly, ORF8 binds CD16a (FcγRIIIA) with nanomolar affinity and decreases the overall level of CD16 at the surface of monocytes and, to a lesser extent, NK cells. This decrease significantly reduces the capacity of PBMCs and particularly monocytes to mediate antibody-dependent cellular cytotoxicity (ADCC). Overall, our data identifies a new immune-evasion activity used by SARS-CoV-2 to escape humoral responses.

## 1. Introduction

Since the discovery of the severe acute respiratory syndrome coronavirus 2 (SARS-CoV-2) in late 2019 in Wuhan, China [1,2,3], major advances have been made in understanding this virus and the disease it causes. A large part of these advances have focused on one SARS-CoV-2 protein, its Spike, which resulted in the development of Spike therapeutic antibodies that have reached the clinic and currently-approved vaccines. However, SARS-CoV-2 has a large and complex genome which codes for 29 proteins [4,5,6] which all play important roles in viral infection, replication, and pathogenesis. Most of these viral proteins remain unfortunately poorly studied compared to the Spike glycoprotein. This includes the small (121 amino acids) and rapidly evolving accessory protein open reading frame 8 (ORF8) [7]. Intriguingly, ORF8 is the most variable accessory protein among SARS-related coronaviruses (SARSr-CoVs) [6,8,9] and, in the initial phase of the pandemic, a deletion in ORF8 was observed [10,11] which led to milder cases of the disease [12]. Of note, a deletion was also observed in the ORF8 of SARS-CoV-1 in 2002–2003 which led to the split into ORF8a and ORF8b [13,14,15]. It is still unclear if those deletions are the result of genomic instability and/or adaptation to the new host. However, in the case of SARS-CoV-1, it has been hypothesized that the 29 nucleotide deletion contributed to zoonotic transition and favored human adaptation [14,15,16], although another hypothesis has also been proposed (i.e., the “founder effect”) [17,18]. SARS-CoV-2 ORF8 has two dimerization interfaces [19] and has been linked to immune evasion. Notably, ORF8 was shown to directly interact with major histocompatibility complex class I molecules (MHC-I) and mediate their downregulation by targeting them to lysosomal degradation via autophagy rendering infected cells more resistant to lysis by cytotoxic T cells [20]. Interestingly, a recent study has shown that ORF8 is secreted from infected cells, can be detected in the plasma of acutely-infected individuals, and is negatively associated with survival [21]. Whether this association is linked to its capacity to induce a cytokine storm remains to be determined [22].

Vaccine-elicited humoral responses were shown to protect from infection and severe disease [23,24]. Anti-SARS-CoV-2 Spike-specific antibodies can mediate a wide range of actions from viral neutralization to different fragment crystallization region (Fc)-mediated effector functions. Among the later, antibody-dependent cellular cytotoxicity (ADCC) and antibody-dependent cellular phagocytosis (ADCP) result in the elimination of infected cells. In the transgenic human ACE2 K18 mice model, Fc-mediated effector functions of neutralizing antibodies have been shown to be required for protection from lethal SARS-CoV-2 challenges [25]. Similarly, strong Fc-mediated effector functions, in the absence of neutralization, were sufficient to significantly delay death [26]. Furthermore, Fc-mediated effector functions were associated with survival in SARS-CoV-2 acutely infected individuals [27,28]. Altogether, these results suggest that Fc-effector functions have a major impact on virus clearance and disease outcome. Accordingly, Fc-effector functions were associated with protection from infection by emerging SARS-CoV-2 variants of concern [29]. These documented inhibitory impacts of Fc-effector functions on virus replication and disease might explain why viruses have developed sophisticated strategies to evade this important immune function [30,31,32,33]. For example, human immunodeficiency virus 1 (HIV-1) uses two accessory proteins, negative regulatory factor (Nef) and viral protein U (Vpu), to protect infected cells from ADCC responses [34,35,36,37]. Other viruses, such as Herpes simplex virus type 1 (HSV-1) and type 2 (HSV-2), murine cytomegalovirus (MCMV), human cytomegalovirus (HCMV) and varicella-zoster virus (VZV) secrete proteins that bind to the Fc portion of host immunoglobulins [38,39,40,41]. These virally encoded Fc binding proteins are thought to contribute to protection from Fc-effector functions. Since acute SARS-CoV-2 infection is associated with a loss of CD16+ cells [42,43], in the present study, we evaluated whether soluble ORF8 could modulate Fc-effector functions.

## 2. Materials and Methods

### 2.1. Ethics Statement

The study was conducted in accordance with the Declaration of Helsinki in terms of informed consent and approval by an appropriate institutional board. Peripheral blood mononuclear cells (PBMCs) and plasmas from naïve, convalescent, and vaccinated individuals were obtained from donors who consented to participate in this research project at CHUM. The protocol was approved by the Ethics Committee of CHUM (protocol #19.381, approved on 25 March 2020). Donors met all eligibility criteria, namely previously confirmed coronavirus disease 2019 (COVID-19) infection and a complete resolution of symptoms for at least 14 days.

### 2.2. Cell Lines and Primary Cells

Human embryonic kidney (HEK)293T cells (obtained from ATCC) were maintained at 37 °C under 5% CO_2_ in Dulbecco’s Modified Eagle Medium (DMEM) (Wisent, St. Bruno, QC, Canada), supplemented with 5% fetal bovine serum (FBS) (VWR, Radnor, PA, USA) and 100 U/mL penicillin/streptomycin (Wisent, St. Bruno, QC, Canada). Human peripheral blood mononuclear cells (PBMCs) obtained by leukapheresis and Ficoll-Paque density gradient isolation were cryopreserved in liquid nitrogen until further use. Monocytes were isolated and separated from resting PBMCs using a human CD14 positive selection kit. The selection was carried out according to the manufacturer protocol (EasySep^TM^ Human CD14 Positive Selection Kit II, STEMCELL Technologies Inc., Vancouver, BC, Canada). PBMCs, monocytes, and PBMCs depleted in monocytes were maintained in complete medium (RPMI 1640 (Thermo Fisher Scientific, Waltham, MA, USA) supplemented with 10% FBS (VWR, Radnor, PA, USA) and 100 U/mL penicillin/streptomycin (Wisent, St. Bruno, QC, Canada)). The CEM.NKr CCR5+ parental cells and CEM.NKr.Spike cells were previously described [44] and maintained in complete RPMI media.

### 2.3. ORF8 Production

Three million HEK293T cells were seeded on 100 mm petri dishes. The following day, the HEK293T cells were transfected with 10 µg of ORF8 DNA (Addgene, Watertown, MA, USA, Cat. 141390) using a standard calcium phosphate protocol. Alternatively, HEK293T cells were seeded in 100 mm petri dishes and transfected with a control plasmid (pcDNA 3.1). After 16 h, the media were changed for both ORF8 transfected and non-ORF8 (pcDNA 3.1) transfected cells. 24 h after the media was changed, the cell supernatant was centrifuged at 484× *g* and aliquoted in 1.5 mL tubes. The tubes were then stored at −80 °C until further use.

### 2.4. ORF8 Western Blot

45 μg lysates of HEK293T cells transfected with SARS-CoV-2 ORF8 DNA and 50 μL of culture supernatants were separated through 12% sodium dodecyl sulfate polyacrylamide gel electrophoresis (SDS-PAGE), then transferred onto the polyvinylidene difluoride (PVDF) membrane. ORF8 protein was probed with sheep anti-ORF8 antibody (dilution 1:1000, MRC PPU Reagents and Services, Dundee, Scotland, Cat. DA088) for 16 h at 4 °C, followed by incubation with donkey anti-sheep horseradish peroxidase (HRP)-conjugated secondary antibody (1:2000, Invitrogen, Waltham, MA, USA, Cat. AP184P) for 1 h at room temperature. The membrane was treated with the Enhanced chemiluminescence (ECL) reagent and protein signals were exposed to the X-ray films.

### 2.5. Staining of PBMCs with ORF8

Five million PBMCs were incubated with Fluorescein isothiocyanate (FITC)-conjugated ORF8 (1 μg) in 100 μL of phosphate-buffered saline (PBS) (containing 3% bovine serum albumin (BSA)) on ice for 30 min. The recombinant ORF8 (Thermo Fisher Scientific, Waltham, MA, USA, Cat. RP-87666) was conjugated with FITC using the FITC conjugation kit (Abcam, Cambridge, UK, Cat. ab102884). Then, the anti-CD14-V450 (dilution 1:100, BD, Franklin Lakes, NJ, USA, Cat. 561390), anti-CD16-PE-Cy7 (dilution 1:100, BD, Franklin Lakes, NJ, USA, Cat. 560716) and anti-CD56-PE (dilution 1:100, BD, Franklin Lakes, NJ, USA, Cat. 556647) antibodies were added for 1 h incubation on ice. Similarly, CD3+ T cells were stained with anti-CD3-PE antibody (dilution 1:100, BD, Franklin Lakes, NJ, USA, Cat. 552127) on ice for 1 h. After washing with PBS (3% BSA), cells were fixed with 4% paraformaldehyde (PFA) for 15 min at room temperature. Cells were then suspended in PBS (3% BSA) and examined with LSP Fortessa flow cytometer. The data were analyzed with FlowJo. The cell gating strategy is presented in Appendix A.

### 2.6. Bio-Layer Interferometry (BLI)

Binding kinetics were performed on an Octet RED96e system (FortéBio, Fremont, CA, USA) at 25 °C with shaking at 1000 rotations per minute (RPM). Amine Reactive Second-Generation (AR2G) biosensors were hydrated in water, then activated for 300 s with an S-NHS/EDC solution (FortéBio, Fremont, CA, USA) prior to amine coupling. CD16a ectodomain (Human CD16a, amino acids Met1-Gln208 (Accession # P08637-1) with a C-terminal His-tag, Thermo Fisher Scientific, Waltham, MA, USA, catalogue number A42538) was loaded into AR2G biosensor at 12.5 µg/mL in 10 mM acetate solution pH 5 (FortéBio, Fremont, CA, USA) for 600 s and then quenched into 1 M ethanolamine solution pH 8.5 (FortéBio, Fremont, CA, USA) for 300 s. Baseline equilibration was collected for 120 s in 10× kinetics buffer. Association of ORF8 protein (SARS-CoV-2 ORF8 (aa16-121), His Tag (RP-87666), Thermo Fisher Scientific, Waltham, MA, USA) (in 10× kinetics buffer) to CD16a ectodomain was carried out for 180 s at various concentrations in a two-fold dilution series from 500 nM to 31.25 nM prior to dissociation for 300 s. The data were baseline subtracted prior to fitting performed using a 1:1 binding model in the FortéBio data analysis software. Calculation of on-rate (K_a_), off-rate (K_dis_), and dissociation constant (K_D_) were computed using a global fit applied to all data.

### 2.7. Co-Immunoprecipitation of CD16a and ORF8

Four million HEK293T cells were seeded on 10 cm dishes. Cells were transfected with either 4 μg empty QCXIP plasmid (Takara Bio Inc., Kusatsu, Japan), 2 μg Flag-tagged CD16a-expressing plasmid (OriGene, Rockville, MD, USA, Cat. RC206429), 2 μg StrepII-tagged ORF8 plasmid, or co-transfected with CD16a and ORF8 DNA, keeping total DNA levels at 4 μg with empty vector. Transfection media was replenished after 16 h. Cells were lysed 48 h post transfection in RIPA buffer (500 μL) containing 50 mM Tris-HCl (pH 7.5), 150 mM NaCl, 1 mM EDTA, 1% Nonidet P40, 0.1% SDS and protease inhibitor (Thermo Fisher Scientific, Waltham, MA, USA, Cat. A32959) for 1 h on ice, and lysates were clarified by centrifugation (13,200× *g* rpm for 15 min at 4 °C). Clarified lysates (10 μg) were mixed with 4× Laemmli buffer, heated at 95 °C for 5 min and analyzed by Western blot as input cell lysates. Co-immunoprecipitation was performed by incubating 1 mg of remaining cell lysates with 15 μL of Anti-Flag M2 Affinity Gel (Sigma-Aldrich, Saint-Louis, MO, USA, A2220-5mL) with rotation for 2 h at 4 °C. Beads were then extensively washed with RIPA buffer and clarified by centrifugation (6000× *g* rpm for 5 min at 4 °C), after which beads were eluted in 1× Laemmli buffer (30 μL) at 95 °C for 3 min. Co-IP samples (5 μL) and cell lysates were analyzed by 12% SDS- polyacrylamide as described above. Membranes were probed with the following primary antibodies: rat anti-Strep-II (1:5000, Abcam, Cambridge, UK, Cat. ab252885), mouse anti-Flag (1:5000, Sigma-Aldrich, Saint-Louis, MO, USA, Cat. F1804-200UG) and mouse anti-tubulin (1:5000, Santa Cruz Biotechnology Inc., Dallas, TX, USA, Cat. SC-23948). They were then probed with the following secondary antibodies conjugated to HRP: goat anti-rat IgG (1:10,000, Invitrogen, Waltham, MA, USA, Cat. 31470) and goat anti-mouse IgG (1:5000, Seracare Life Sciences Inc., Milford, MA, USA, Cat. 5450-0011). The membranes were then imaged as described above.

### 2.8. CD16 Cell-Surface Staining on Monocytes or Natural Killer (NK) Cells

PBMCs, purified monocytes, or PBMCs depleted in monocytes from different donors were treated with ORF8 produced in 293T cells for 16 h at 37 °C and 5% CO_2_. As control, PBMCs were treated with an identical volume of conditioned media collected from 293T cells transfected with a control plasmid (pc DNA 3.1). Alternatively, PBMCs were treated with a commercially available ORF8 (SARS-CoV-2 ORF8 (aa16-121), His Tag (RP-87666), Thermo Fisher Scientific, Waltham, MA, USA) at different concentrations (from 3.2 pg/10^6^ PBMCs to 250 ng/10^6^ PBMCs). Following treatment, cells were stained with anti-CD3 BV-421 (BioLegend, San Diego, CA, USA, Cat. 300434), anti-CD19 BV-421 (BioLegend, San Diego, CA, USA, Cat. 302233), anti-CD14 PerCP-Cy5.5 (BD, Franklin Lakes, NJ, USA, Cat. 550787), anti-CD56 PE (BD, Franklin Lakes, NJ, USA, Cat. 340363), anti-CD16 FITC (BD, Franklin Lakes, NJ, USA, Cat. 555406) and LIVE/DEAD Fixable Aqua Dead Cell Stain (Thermo Fisher Scientific, Waltham, MA, USA, Cat. L34966) for 25 min at 4 °C to identify the monocytes and NK cells among the PBMC population and to measure their surface level of CD16. After staining, the cells were fixed with 2% PFA and stored at 4 °C until the samples were acquired on a LSRII cytometer (BD Biosciences, Mississauga, ON, Canada). Data analysis was performed using FlowJo v10.5.3 (Tree Star, Ashland, OR, USA).

### 2.9. Antibody-Dependent Cellular Cytotoxicity (ADCC) Assay

This assay was previously described [44,45]. Briefly, for evaluation of anti-SARS-CoV-2 ADCC activity, parental CEM.NKr CCR5+ cells were mixed at a 1:1 ratio with CEM.NKr.Spike cells. These cells were stained for viability (AquaVivid; Thermo Fisher Scientific, Waltham, MA, USA) and a cellular dye (cell proliferation dye eFluor670; Thermo Fisher Scientific, Waltham, MA, USA) and subsequently used as target cells. PBMCs, purified monocytes or PBMCs depleted in monocytes were treated with ORF8 which was produced in 293T cells or were treated with an identical volume of conditioned media collected from 293T cells transfected with a control plasmid (pc DNA 3.1). Alternatively, PBMCs were treated with ORF8 at 50 ng/10^6^ PBMCs (SARS-CoV-2 ORF8 (aa16-121), His Tag (RP-87666), Thermo Fisher Scientific, Waltham, MA, USA). Treatment was carried out overnight for 16 h at 37 °C and 5% CO_2_. These cells were stained with another cellular marker (cell proliferation dye eFluor450; Thermo Fisher Scientific, Waltham, MA, USA) and used as effector cells. Stained effector and target cells were mixed at a 10:1 ratio in 96-well V-bottom plates. Plasma from convalescent, vaccinated or convalescent and vaccinated individuals at a dilution of 1/500 was added to the appropriate wells. The plates were subsequently centrifuged for 1 min at 300× *g*, and incubated at 37 °C, 5% CO_2_ for 5 h before being fixed in a 2% PFA solution. Since CEM.NKr.Spike cells express green fluorescent proteins (GFP), ADCC activity was calculated using the formula: [(% of GFP + cells in Targets plus Effectors) − (% of GFP + cells in Targets plus Effectors plus plasma)]/(% of GFP + cells in Targets) × 100 by gating on transduced live target cells. All samples were acquired on an LSRII cytometer (BD Biosciences, Mississauga, ON, Canada) and data analysis performed using FlowJo v10.5.3 (Tree Star, Ashland, OR, USA).

### 2.10. Statistical Analyses

Statistics were analyzed using GraphPad Prism version 8.0.2 (GraphPad, San Diego, CA, USA). Every data set was tested for statistical normality and this information was used to apply the appropriate (parametric or nonparametric) statistical test. *p* values < 0.05 were considered significant (non-significant values indicated as n.s.); significant values are indicated as * *p* < 0.05, ** *p* < 0.01, *** *p* < 0.001 and **** *p* < 0.0001.

## 3. Results and Discussion

### 3.1. SARS-CoV-2 ORF8 Interacts with CD16a

We first evaluated whether ORF8 could interact with different cell types present in PBMCs. To investigate this possibility, recombinant ORF8 was conjugated with FITC and incubated on ice with PBMCs for 30 min. Ten percent of cells present in PBMCs were bound by ORF8 (Figure 1A). By further investigating which cell type bound ORF8 using anti-CD14, anti-CD16 and anti-CD56 antibodies, we found that ORF8 bound monocytes (CD14+CD16+) as well as NK cells (CD56+CD16+), while weakly binding with CD3+ T cells (Figure 1A). Since it has been previously shown that ORF8 adopts an Ig-like fold ^19^ and since CD16 is a common receptor at the surface of monocytes and NK cells, we asked whether ORF8 could directly interact with CD16a. Using bio-layer interferometry (BLI), we measured the affinity of recombinant ORF8 with the CD16a ectodomain. As presented in Figure 1B, ORF8 bound CD16a with nanomolar affinity (38.9 nM, Figure 1B). The interaction of ORF8 with CD16a was further verified by the results of co-immunoprecipitation performed with lysates of HEK293T cells that we co-transfected with CD16a and ORF8 expression vectors (Figure 1C).

### 3.2. SARS-CoV-2 ORF8 Decreases CD16 Levels at the Surface of Monocytes and NK Cells 

CD16 interacts with the Fc portion of IgGs and is present at the surface of different cell types including NK cells and monocytes. Upon binding a sufficient amount of antibodies, these cells get activated and can mediate ADCC. To evaluate whether ORF8 could modulate this response, we first assessed whether ORF8 modulated CD16 levels at the surface of NK cells and monocytes. Briefly, PBMCs were incubated overnight with or without ORF8 as described in material and methods. Sixteen hours after ORF8 treatment, PBMCs were stained with anti-CD3, anti-CD14, anti-CD19, anti-CD56 and anti-CD16 antibodies and analyzed by flow cytometry to measure CD16 levels at the surface of NK cells and monocytes. As presented in Figure 1D, the surface level of CD16 on monocytes present within the PBMC population was significantly decreased upon treatment with ORF8. When looking at CD16 at the surface of NK cells, a smaller, but significant, decrease was also observed (Figure 1E). Using a commercially available recombinant ORF8 protein, we recapitulated this phenotype and further show that the phenotype is dose dependent Appendix A. ORF8-induced decrease of cell surface CD16 started at a concentration of 3.2 pg/10^6^ PBMCs before stabilizing at around 2 ng/10^6^ PBMCs (Appendix A). To evaluate if ORF8 directly acted on monocytes to decrease their CD16 level, we used monocytes purified from PBMCs and incubated them with ORF8 produced in HEK293T cells. As shown in Appendix A, addition of soluble ORF8 to purified monocytes modulated CD16 levels in a manner similar to when added to total PBMCs, therefore suggesting a direct effect of ORF8 on monocytes. To verify if the decreased CD16 levels at the surface of NK cells also depended on the presence of monocytes, ORF8 was added to monocyte-depleted PBMCs. As shown in Appendix A, the small decrease on NK cells was not observed upon monocyte depletion, confirming the role of monocytes in ORF8-mediated downmodulation of CD16 on NK cells. These results suggest that monocytes represent an important target of ORF8.

### 3.3. SARS-CoV-2 ORF8 Decreases ADCC Responses Mediated by PBMCs and Monocytes

Since NK cells and monocytes mediate ADCC by directly engaging the Fc region of IgGs via CD16 [46,47], we next evaluated if the observed decrease of CD16 at the surface of monocytes and NK cells affected ADCC responses with an in vitro assay which uses PBMCs as effector cells [44,45]. We first produced soluble ORF8 in culture media by transfecting HEK293T cells with ORF8 DNA. In agreement with previous data showing ORF8 secretion [21,22], we also observed ORF8 secretion and supernatant accumulation (Figure 2A). PBMCs were treated overnight with ORF8-conditioned media or conditioned media alone before being used as effector cells in the ADCC assay. Alternatively, PBMCs were treated overnight with a recombinant ORF8 protein at a concentration of 50 ng/10^6^ PBMCs. In agreement with ORF8-mediated CD16 downregulation, soluble ORF8 present in conditioned media as well as recombinant ORF8 significantly decreased ADCC responses mediated by plasma from convalescent and vaccinated individuals (Figure 2B–D, Appendix A). The contribution of monocytes in these results was assessed using ORF8-treated monocytes. In agreement with decreased CD16 levels at the surface of ORF8-treated monocytes, we observed a significant decline in monocyte-mediated ADCC (Figure 2E). Of note, the decrease in CD16 levels and ADCC responses after the addition of ORF8 to total PBMCs or purified monocytes was similar (Appendix A), suggesting that the impact of ORF8 on PBMCs is mostly dependent on monocytes. As expected, plasma from uninfected/unvaccinated individuals did not mediate ADCC using PBMCs as effector cells and ORF8 did not significantly impact this response (Appendix A).

Viruses employ a wide range of strategies to evade host immune responses and SARS-CoV-2 makes no exception. For example, the SARS-CoV-2 Spike mutates at a fast rate which results in the apparition of multiple variants [48]. These variants have a decreased recognition of their Spike by antibodies from previous infections and/or vaccination. As a result, a decreased neutralization capacity has been observed against some of these variants [49]. Another immune evasion strategy used by SARS-CoV-2 is to downregulate MHC-I at the surface of infected cells, rendering them more resistant to lysis by cytotoxic T cells. The ORF8 protein was found to interact with MHC-I molecules and mediates their downregulation [20]. However, while MHC-I downregulation decreases infected cells susceptibility to the cytotoxic T cell response, this could potentially result in enhanced susceptibility to NK cell responses. NK cell effector functions are controlled by an array of inhibitory and activating receptors that recognize multiple ligands present at the surface of target cells [50]. MHC-I acts as ligands of NK cell inhibitory receptors, including the Killer Ig-like receptors (KIR). Downregulation of MHC-I molecules could theoretically shift the balance toward the activation of NK cell effector functions. To counteract this potential increase in susceptibility to NK cell responses, SARS-CoV-2 might have developed other strategies to evade these responses. Another immune response playing an important role against SARS-CoV-2 infection has been shown to be Fc-mediated effector functions which include ACDP and ADCC [25,26]. Furthermore, high ADCC responses have been associated with survival [27]. Here, we show that ORF8 decreases ADCC responses likely as a result of reduced CD16 from the surface of monocytes and NK cells. However, whether this phenotype is associated with the ability of ORF8 to directly interact with CD16 remains to be validated. The impact of secreted ORF8 on CD16 could potentially compensate for the downmodulation of MHC-I molecules and therefore prevent the elimination of infected cells by NK cells. Our findings support the idea that ORF8 represents a major player in SARS-CoV-2-mediated immune evasion by counteracting immune responses mediated by NK cells and monocytes. 

In a previous study, it was demonstrated that in K18-hACE2 mice, monocytes and NK cells were important for the success of SARS-CoV-2 neutralizing antibody-directed therapy [25]. Indeed, when monocytes or NK cells were depleted from the mice, 80% and 25% of the mice died, respectively, whereas 100% of the mice survived the lethal SARS-CoV-2 infection when no depletion took place. Interestingly, a more pronounced effect was observed on the decrease of CD16 on monocytes compared to NK cells [25]. Nevertheless, the precise mechanism behind the downmodulation of surface CD16 on monocytes and NK cells remains unclear. One possibility might be that upon ORF8 binding, CD16 sheds from the cell surface. Alternatively, the ORF8-CD16 interaction could potentially induce the secretion of one or many cytokines and/or a signalization cascade resulting in the endocytosis of CD16 and its loss from the cell surface. Further investigation will be required to identify the precise mechanism behind this novel activity of ORF8. 

Vaccine-elicited humoral responses are associated with vaccine efficacy against SARS-CoV-2 infection and protection from severe disease [23,24]. In response to the antibodies produced by vaccinated and already infected individuals, multiple variants are emerging [48]. A common feature of these variants is the apparition of multiple mutations in the Spike. These mutations decrease antibody recognition which cause immune evasion and increased transmissibility [48]. Mutations were also observed in ORF8 but their effect(s) on immune evasion remain largely unknown. Our results raise the intriguing possibility that emerging variants use ORF8 to evade Fc-effector functions by impairing monocytes and to a lesser extent NK cells to mediate ADCC.

## Figures and Tables

**Figure 1 viruses-14-01237-f001:**
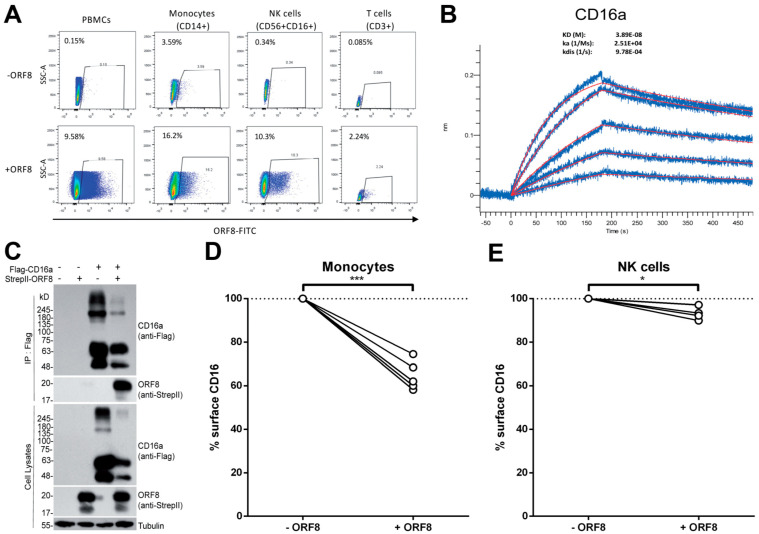
**ORF8 binds monocytes and NK cells through CD16a.** (**A**) PMBCs were incubated with or without FITC-conjugated recombinant ORF8 protein on ice for 30 min, followed by staining with anti-CD14-V450, anti-CD16-PE-CY7 and anti-CD56-PE antibodies before being fixed with 4% PFA and analyzed by flow cytometry. CD3+ T cells were labeled with an anti-CD3-PE antibody. (**B**) AR2G biosensors loaded with CD16a protein were soaked in two-fold dilution series of ORF8 (31.25 nM–500 nM). Raw data are shown in *blue* and model in *red*. The dissociation/affinity constant (K_D_), on rates (K_a_) and off rates (K_dis_) were calculated using a 1:1 binding model. (**C**) HEK293T cells were transfected with CD16a and ORF8 plasmid DNA, followed by co-immunoprecipitation with an anti-Flag antibody. The presence of ORF8 was detected with the anti-StrepII antibody. (**D**,**E**) PBMCs from different donors were thawed and incubated for 16 h with media from ORF8 DNA-transfected HEK293T cells. Alternatively, PBMCs were treated with an identical volume of conditioned media collected from HEK293T cells transfected with a control plasmid. The following day, the PBMCs were stained with anti-CD3, anti-CD14, anti-CD19, anti-CD56, anti-CD16 and LIVE/DEAD Fixable Aqua Dead Cell Stain and analyzed by flow cytometry to measure surface levels of CD16 on (**D**) monocytes (*n* = 5) and (**E**) NK cells (*n* = 4). Cell-surface CD16 levels in presence of ORF8 were normalized on cell-surface CD16 detected in absence of ORF8. Statistical significance was evaluated using a parametric paired *t*-test. *, *p* < 0.05; ***, *p* < 0.001.

**Figure 2 viruses-14-01237-f002:**
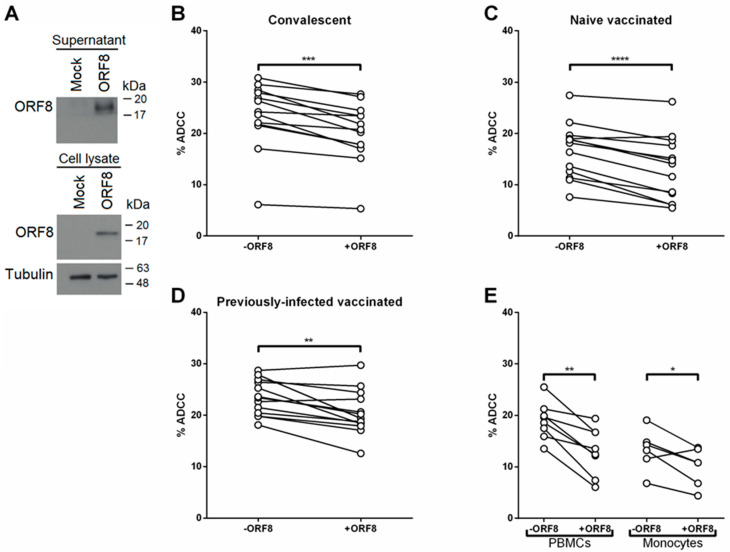
**ORF8 decreases PBMC-mediated ADCC.** (**A**) Secretion of ORF8 into culture supernatant of HEK293T cells transfected with ORF8 DNA. ORF8 was probed with an anti-ORF8 antibody. (**B**) ADCC (%) mediated by plasma from 13 convalescent individuals, (**C**) 13 vaccinated individuals, and (**D**) 13 previously-infected vaccinated individuals using PBMCs from healthy donors as effector cells. (**E**) ADCC (%) mediated by plasma from convalescent individuals with PBMCs or monocytes as effector cells. The effector cells were treated overnight (16 h) with media from ORF8 DNA-transfected HEK293T cells (+ORF8) or with an identical volume of conditioned media collected from HEK293T cells transfected with a control plasmid (−ORF8). *, *p* < 0.05; **, *p* < 0.01; ***, *p* < 0.001; ****, *p* < 0.0001.

## Data Availability

The data are contained within the article and Appendix A.

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
