# Peer review of "SARS-CoV-2 Accessory Protein ORF8 Decreases Antibody-Dependent Cellular Cytotoxicity"

_viruses, 2022, doi:10.3390/v14061237_

Round 1

Reviewer 1 Report

Comments:

In Figure 1A. Please include the gating strategy as supplementary figure. Please also include the plot of CD14 on Monocyte (without orf8) and CD56+; CD16+ NK cells (without orf8) in the main figure.

Figure 1A included the flow data of PBMC, Monocyte and NK. Please also include the flow data of other immune cell types that express relatively low level of CD16 for comparison. This can help to demonstrate the specificity and significance of the binding of orf8 and CD16a.

For Figure 1C and D and Figure 2, the major concern is the purity of the orf8 protein for the incubation experiment. I cannot find any purification step in the method 2.3, to get rid of the serum and other components from the culture medium. The use of non-purified orf8 protein, or the 16hr treatment of cells with the impurity (in conditioned media) such as serum may activate immune cells non-specifically and hence affect the conclusion of the experiments. It is recommended to repeat the experiments with purified protein and proper controls.

Overall, the binding of orf8 to CD16 is interesting. The functional assays need improvement.

Reviewer 2 Report

The research work by Beaudoin-Bussières and colleagues, entitled "SARS-CoV-2 accessory protein ORF8 decreases antibody-dependent
cellular cytotoxicity" presents in the form of a short communication results describing a novel mechanism exerted by SARS-CoV-2 ORF8 protein to escape host humeral immunity, namely by binding to CD16 receptor, thereby decreasing its exposure at the surface of monocytes and NK cells and, ultimately, reducing the antibody-dependent cellular cytoxicity response.

The research is well presented, methodologically correct and scientifically sound and will be of great interest for the readership of the journal Viruses. Therefore, it is opinion of this reviewer that this work should be accepted with minor revision of the manuscript, particularly in its introductory part (abstract and introduction), as well as throughout the manuscript text to check for correctness and lack of acronyms explicitation.

Two major suggestions are made by this reviewer to encourage the authors in highlighting the quality of their work:

1) The abstract first lines suffer from an overemphasis on the properties of the Spike protein, while the main target of the study, the ORF8 protein, is only mentioned after five lines.

2) Although the introduction is reasonably well written, especially considering the word count constraints in the context of a short communication, the authors should put more efforts in explaining - to an hypothetical readership that is not familiar with the molecular biology and natural history of SARS-related coronaviruses - why ORF8 deletion is supposed to occur (e.g. as a putative process of host adaptation under evolutionary pressure), that such deletions were observed in the initial phase of the Covid-19 pandemic as well as, for SARS-CoV ORF8, similar ones in the early phase of the 2002-2003 outbreak, and so on. Even without going into much detail, authors may easily summarise the background in one phrase and, possibly, refer to one of the few exhaustive recent reviews on ORF8 that are available in the scientific literature.

A general, minor recommendation to the authors, following which in my opinion they would significantly improve the quality of their work, is to consistently define all acronyms throughout the text, first mentioning the explicit and extended form of the name, and then its conventional acronym in brackets. And from that line onward, only using the acronym. Please, refer to the following minor suggestion list for this: 

Line 37-41: its Spike (S)... development of S therapeutic....compared to the S glycoprotein.
Line 39: coding for up to 29-30 proteins
Line 42: open reading frame 8 (ORF8)..,
Line 47: [...] can be (remove space)
Line 52: S-specific
Line 53: fragment crystallisation region (Fc)-mediated...
Line 57: SARS-CoV-2 challenges
Line 60: SARS-CoV-2 acutely infected individuals...
Line 65: human immunodeficiency virus 1 (HIV-1)...
Line 66: negative factor (Nef) and viral protein U (Vpu), ...

Line 71: please, the authors are encouraged to add a reference for CD16+ cell loss during SARS-CoV-2 acute infection. Moreover, in order to make their research work more interesting and appealing to the broadest readership, this reviewer invites the authors to pur some more effort here in introducing the cluster of differentiation 16 (CD16) as Fc-receptor on the surface of monocytes, macrophages, natural killer (NK) cells, etc. (maybe just in the same way - by moving the text here - as they have done later on, at lines 185-187). Also in this regard, consider that the other name, Fc-gamma RIIIa, was only mentioned in the abstract and as title in one figure panel (B of Fig.1), whereas it was never introduced in the text.

Line 82: coronavirus disease 2019 (COVID-19) infection...

Line 84: Human embryonic kidney (HEK) 293T cells
Line 87: human PBMCs
Line 92: monocytes
Line 94: FBS
Line 99: SARS-CoV-2 ORF8 DNA (GenBank code: xxxxxx)

Line 103: ORF8 Western blot
Line 104: with SARS-CoV-2 ORF8 DNA
Line 105: sodium dodecyl phosphate polyacrylamide gel electrophoresis (SDS-PAGE)
Lines 106-111: please, place acronyms (i.e. PVDF, HRP, ECL) into brackets and their explicit definitions out of brackets.

Line 113:fluorescein isothiocyanate (FICT)-conjugated... phosphate-buffered saline (PBS)
Line 114: bovine serum albumin (BSA)
Line 120: flow cytometer (add brand)
Line 121: FlowJo software (add brand or url website: maybe as already done at lines 145 and 164?)
Lines 127-133: please, check and review all-opened brackets
Line 136: affinity dissociation (Kd) constants, were computed
Line 137: natural killer (NK) cells
Line 143: paraformaldehyde (PFA)
Line 160: green fluorescent protein (GFP)
Line 172: SARS-CoV-2 ORF8 interacts...
Line 174: present in PBMCs
Line 184: SARS-CoV-2 ORF8 decreases...
Line 212: Kd
Line 219: SARS-CoV-2 ORF8...
Line 245: S protein

Reviewer 3 Report

  1. Fig1A, The experiments that ORF8 bound monocytes (CD14+CD16+) and NK cells (CD56+CD16+) require negative controls.

  1. Fig1B, It would be better to validate the interaction of ORF8 and CD16a by immunoprecipitation or pull-down assay.

  1. Fig1C and 1D, If ORF8 decreases surface levels of CD16 can be proved in stable cell lines, the conclusion will be more reliable.

  1. The authors could try to discuss how ORF8 reduces ADCC responses and the role of monocytes in this process.

Round 2

Reviewer 1 Report

Comments:

My previous comment on Figure 1A “Please also include the flow data of other immune cell types that express relatively low level of CD16 for comparison.”

In this revised manuscript, authors added the CD3+ cells (presuming CD16 is low) only. First, why authors did not include more immune cell types that express relatively low level of CD16 for comparison. I believe authors have already investigated the major immune cell-types in PBMC during this study and hence concluded monocyte is the key player. It is recommended to include the complete set of data (i.e. including other immune cell types as recommended in the first review, but not CD3+ cells only) so that the readers will know the overall picture. Second, why author didn’t plot the ORF8-FITC and CD16-PE-Cy7 in the same plot? The double positive cells (ORF8 hi and CD16 hi) will strengthen your conclusion. Please improve your presentation.

My previous comment “The binding of orf8 to CD16 is interesting. The functional assays need improvement.” Authors use plasma from convalescent and vaccinated individuals for ADCC assay. The assays should include negative control, such as the use of non-infected or non-vaccinated plasma.

Co-IP data further supports the binding of ORF8 with CD16.

Addition of recombinant ORF8 experiment further supports the specificity.

Author Response

My previous comment on Figure 1A “Please also include the flow data of other immune cell types that express relatively low level of CD16 for comparison.”

In this revised manuscript, authors added the CD3+ cells (presuming CD16 is low) only. First, why authors did not include more immune cell types that express relatively low level of CD16 for comparison. I believe authors have already investigated the major immune cell-types in PBMC during this study and hence concluded monocyte is the key player. It is recommended to include the complete set of data (i.e. including other immune cell types as recommended in the first review, but not CD3+ cells only) so that the readers will know the overall picture. Second, why author didn’t plot the ORF8-FITC and CD16-PE-Cy7 in the same plot? The double positive cells (ORF8 hi and CD16 hi) will strengthen your conclusion. Please improve your presentation.

Response: We thank the reviewer for raising this point. We  do agree with this reviewer that it would be insightful to perform a comprehensive examination of ORF8 binding to all immune cell subsets in PBMCs, as he/she suggested.  This is something that we plan to do by selecting a panel of antibodies conjugated with compatible fluorophores that can be used together to label most of the immune cell types.  Unfortunately, we have not done this experiment yet. In the meantime, we believe that our current data support substantial binding of ORF8 to monocytes and NK cells (the focus of this manuscript), as compared to CD3+ T cells, and are in agreement with the adverse effect of ORF8 on ADCC which is mainly performed by monocytes and NK cells in PBMCs. We specifically gated the CD56+CD16+ NK cells for analysis because they are the main effector cells for ADCC, whereas gating CD16 alone will lead to selection of mixed cell types, which may complicate the interpretation of the data.

My previous comment “The binding of orf8 to CD16 is interesting. The functional assays need improvement.” Authors use plasma from convalescent and vaccinated individuals for ADCC assay. The assays should include negative control, such as the use of non-infected or non-vaccinated plasma.

Response: we have done and reported on several occasions the specificity of our assay using plasma from non-infected, non-vaccinated individuals; we refer to this work in the method section of the manuscript (Beaudoin et al., STAR protocol 2021; Tauzin et al., Cell Host and Microbe 2021).  To avoid overcrowding this “communication” manuscript we prefer to keep this figure as it is.

Co-IP data further supports the binding of ORF8 with CD16.

Response: we thank the reviewer for asking this experiment.  We agree with the reviewer that it further supports and validates our findings.

Addition of recombinant ORF8 experiment further supports the specificity.

Response: we thank the reviewer for requesting this experiment that further supports the conclusions of our manuscript.

Reviewer 3 Report

The authors have addressed all issues raised by the reviewers and editor in the revision of the manuscript.

Author Response

The authors have addressed all issues raised by the reviewers and editor in the revision of the manuscript.

Response: we thank the reviewer for his/her insightful comments that helped strength our manuscript.